# Durable Fast Charging of Lithium-Ion Batteries Based on Simulations with an Electrode Equivalent Circuit Model

Robin Drees [1,2,*] , Frank Lienesch [2,3] and Michael Kurrat [1,2]

1    elenia Institute for High Voltage Technology and Power Systems, Technische Universität Braunschweig, Schleinitzstraße 23, 38106 Braunschweig, Germany; m.kurrat@tu-braunschweig.de

2    Battery LabFactory Braunschweig, Technische Universität Braunschweig, Langer Kamp 19, 38106 Braunschweig, Germany; frank.lienesch@ptb.de

3    Physikalisch-Technische Bundesanstalt, Bundesallee 100, 38116 Braunschweig, Germany

*    Correspondence: r.drees@tu-bs.de; Tel.: +49-531-391-9730

**Abstract:** Fast charging of lithium-ion batteries is often related to accelerated cell degradation due to lithium-plating on the negative electrode. In this contribution, an advanced electrode equivalent circuit model is used in order to simulate fast-charging strategies without lithium-plating. A novel parameterization approach based on 3-electrode cell measurements is developed, which enables precise simulation fidelity. An optimized fast-charging strategy without evoking lithium-plating was simulated that lasted about 29 min for a 0–80% state of charge. This variable current strategy was compared in experiments to a conventional constant-current–constant-voltage fast-charging strategy that lasted 20 min. The experiments showed that the optimized strategy prevented lithium-plating and led to a 2% capacity fade every 100 fast-charging cycles. In contrast, the conventional strategy led to lithium-plating, about 20% capacity fade after 100 fast-charging cycles and the fast-charging duration extended from 20 min to over 30 min due to increased cell resistances. The duration of the optimized fast charging was constant at 29 min, even after 300 cycles. The developed methods are suitable to be applied for any given lithium-ion battery configuration in order to determine the maximum fast-charging capability while ensuring safe and durable cycling conditions.

**Keywords:** battery; charging; plating; aging; model; simulation

## 1. Introduction

Improving the fast-charging capability is one of the main challenges in the research of lithium-ion batteries. The main limiting factor of the charging speed is the maximum allowed cell voltage, as the current has to be reduced as soon as the upper voltage limit is reached. High currents or internal resistances lead to higher overpotentials due to Ohm's Law and result in shorter durations until the upper voltage limit is reached. The maximum charging rates of batteries are related to various parameters that influence the overpotential due to internal resistances, such as electrode and electrolyte properties or cell configuration. For example, a comparatively higher electrode porosity leads to also higher surface area between particles and electrolytes, which correlates with lower internal resistance. The ionic conductivity of the electrolyte also influences the internal resistance and differs based on the salt type and concentration [1]. With regard to the cell configuration, there are different electrode materials (cell chemistries) with different electrical conductivities. Based on all these possible variants, cell types are generally classified by either high energy, high power or hybrid demands [2]. Electric vehicles have high energy, as well as high power demands. At the moment, the maximum specified fast-charging rates of lithium-ion batteries for electric vehicles are generally below 3C [3]. In the literature, there are different definitions of 'fast charging' for electric vehicles. According to the California Air Resources Board (ARB), fast charging is defined as 100 miles driving range within 10 min charging [4]. The U.S. Advanced Battery Consortium defines fast charging as at least 40% state of charge

(SOC) within 15 min [5]. There are also other definitions in the literature where charging durations of more than 2 h are classified as 'slow charging', whereas 0.5–2 h are defined as 'fast charging' and even shorter charging times are termed as 'extreme fast charging' [3,6]. Low temperatures lead to longer charging times, whereas higher temperatures lead to better conductive properties that are necessary for extreme charging scenarios [7]. However, critical temperatures due to self-heating are problematic and have to be avoided by external cooling [8].

Extreme high charging rates do not decrease the charging time if the maximum cell voltage is reached immediately. In addition, charging currents that are too high may lead to degradation effects. The most important degradation effect associated with fast charging is called lithium-plating [9,10]. If the voltage of the negative electrode drops below 0 V vs. lithium during charging, metallic lithium is likely to form on the negative electrode's surface [11,12]. This phenomenon results due to the conditions being more energetically favorable compared to lithium intercalation [13–15]. Only a part of the plated lithium is reversibly stripped, which is reusable for further cycling [16]. The non-reversible part leads to capacity fade, undesired side reactions or an internal short circuit in the worst case as lithium-plating can form sharp dendrites [3]. These consequences facilitate earlier cell defects and result in reduced cyclability, as well as a shorter lifetime. Increasing the lithium-ion transport rate by special materials [17–20] may decrease the possibility of lithium-plating due to lower overpotentials.

In order to check batteries for lithium-plating, battery cells need to be disassembled and optical post-mortem analysis of the negative electrodes has to be conducted. As lithium is highly reactive, the literature recommends the cell opening in a glove-box under argon atmosphere at 0% SOC [21]. Periodic cycling with too high currents leads to lithium-plating that is clearly visible when the cell is opened and the negative electrode is inspected under argon atmosphere [22]. There are also investigations that aim to detect lithium-plating by differential voltage analysis (DVA) or analyzing the voltage curves after charging [23]. DVA can also be used in order to estimate the quantity of stripped lithium [24]. The voltage relaxation curve after a charge cycle that has evoked lithium-plating correlates with a voltage plateau [25], which was also confirmed by glow discharge optical emission spectroscopy (GD-OES) depth profiling [26]. Another approach of lithium-plating detection aims at calculating the change of the internal resistance during fast charging, which correlates with the onset of lithium-plating [27]. Special batteries with a lithium reference electrode (3-electrode cells) enable the voltage prediction of the negative electrode during cycling conditions in order to detect and quantify lithium-plating [28,29].

Different simulation models for optimizing fast-charging strategies were developed in order to prevent lithium-plating [3,30]. Most of them are electrochemical models based on Doyle, Fuller and Newman [31]. Based on this approach, there are variants with lumped parameters that focus on predicting the end-of-discharge or end-of-life scenarios [32,33]. Electrochemical models are also appropriate for calculating the positive electrode's and negative electrode's electrochemical behavior or lithium-plating side reactions [34]. This approach was used for models based on electrochemical equations and constraints in order to maximize the current [35] or to control a safe battery state [36] during fast charging. Similar models with control loops for the negative electrode overpotential were developed that downsize the current after a specific minimum voltage of the negative electrode is detected [37,38]. Such optimized current profiles were evaluated by 3-electrode cells and post-mortem analyses and showed reduced or no lithium-plating [37–39]. As lithium-plating is not the only degradation effect in battery cells, there are also fast-charging strategies focusing on other degradation effects, such as minimizing solid electrolyte interphase (SEI) growth [40] or particle stress [41]. Another electrochemical model considered various degradation effects and computed a multi-stage constant current strategy with a stepwise current profile [42].

In general, most electrochemical models require many parameters that are difficult to determine and need to be estimated [43]. Electrochemical models are also challenging for

precise validation, whereas equivalent circuit models (ECMs) have fewer parameters that need to be estimated and therefore are easier to validate [3]. Common ECMs consist of an open circuit voltage source, resistances and capacities and have been used for maximizing charging efficiency [44–46] or limiting heat generation during charging [47,48]. However, common ECMs are not able to consider internal battery state information as electrode potentials or degradation effects. Advanced physics-based ECMs were developed, in order to consider the internal electrode properties [3]. There are physically meaningful ECMs that include local electrode state information, in order to detect degradation mechanisms [49,50]. A similar physics-based ECM was developed that needs three different experimental datasets of discharge tests and impedance spectroscopy under load for the parameterization [51].

In a preliminary study, we developed a novel physics-based ECM—the Electrode Equivalent Circuit Model (EECM). This model is suitable to optimize fast-charging strategies by optimizing the potential of the electrodes. The model requires just a few parameters that are extracted from experimental cycle tests with 3-electrode cells. The EECM was used for designing fast-charging SEI formation strategies in the context of faster battery production [52].

In the present study, the EECM approach is further advanced and parameterized by a new method, in order to optimize a durable fast-charging strategy for cycling without lithium-plating. In Section 2, the advanced EECM and new parameterization method is presented that enables high simulation fidelity. In Section 3, the optimized fast-charging strategy based on the EECM is designed and compared to a common constant-current-constant-voltage (CCCV) fast-charging strategy. Afterward, these strategies are applied to the cycling of battery cells which are then assessed by means of post-mortem, capacity fade and internal resistance analyses.

## 2. Material and Methods

### 2.1. Advanced Electrode Equivalent Circuit Model

For this study, we focused on the model-based design of a durable fast-charging strategy without evoking lithium-plating during cycling. Therefore, the applied fast-charging optimization approach based on the EECM is illustrated in Figure 1.

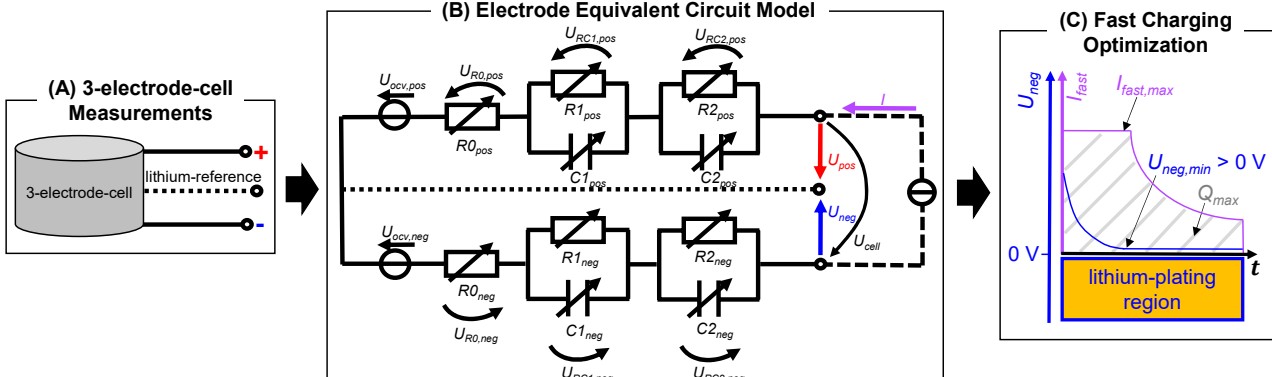

**Figure 1.** Fast-charging optimization approach based on an Electrode Equivalent Circuit Model (EECM) which is parameterized by 3-electrode cell measurements. (**A**) Measurements with 3-electrode cells are the input for the EECM parameterization (details in Section 2.2). (**B**) EECM with an open circuit voltage source ($U_{ocv}$), ohmic resistance ($R0$) and two $RC$ branches ($R1$, $C1$ and $R2$, $C2$) per electrode. (**C**) The control of the voltage of the negative electrode ($U_{neg}$) to non-negative values allows the optimization of fast-charging current profiles ($I_{fast}$) without lithium-plating. The model approach is based on Drees et al. [52] and is further advanced to two $RC$ branches, in order to increase the simulation fidelity.

Every *RC* branch has a specific time constant, which is the product of *R* and *C*, respectively. Two *RC* branches per electrode are more precise than one, as the charge transfer and diffusion processes, that take place in the electrodes during charging, have different time constants [53]. Compared to only one *RC* branch per electrode [52], the model with two *RC* branches per electrode has just a slightly increased parameterization expense but promises increased simulation fidelity.

The model is based on the following equations. Equation (1) computes the cell voltage ($U_{cell}$), which is the difference between the voltage of the positive electrode ($U_{pos}$) and a negative electrode ($U_{neg}$). Equations (2) and (3) represent the electrode voltages based on Ohm's Law and Kirchhoff's Second Law. Equations (4)–(7) are the differential equations for the *RC* branches that calculate the non-ohmic overpotentials.

$$U_{cell}(t) = U_{pos}(t) - U_{neg}(t) \tag{1}$$

$$U_{pos}(t) = U_{ocv,pos} + I(t) * R0_{pos} + U_{RC1,pos}(t) + U_{RC2,pos}(t) \tag{2}$$

$$U_{neg}(t) = U_{ocv,neg} - I(t) * R0_{neg} - U_{RC1,neg}(t) - U_{RC2,neg}(t) \tag{3}$$

$$U_{RC1,pos}(t) = 1/\left(R1_{pos} * C1_{pos}\right) * \int \left(I(t) * R1_{pos} - U_{RC1,pos}(t)\right) dt \tag{4}$$

$$U_{RC2,pos}(t) = 1/\left(R2_{pos} * C2_{pos}\right) * \int \left(I(t) * R2_{pos} - U_{RC2,pos}(t)\right) dt \tag{5}$$

$$U_{RC1,neg}(t) = 1/\left(R1_{neg} * C1_{neg}\right) * \int \left(I(t) * R1_{neg} - U_{RC1,neg}(t)\right) dt \tag{6}$$

$$U_{RC2,neg}(t) = 1/\left(R2_{neg} * C2_{neg}\right) * \int \left(I(t) * R2_{neg} - U_{RC2,neg}(t)\right) dt \tag{7}$$

In contrast to common ECMs without electrode separation, the EECM is able to compute optimized current profiles without lithium-plating by considering the negative electrode's voltage. The objective function by Equation (8) maximizes the current in order to compute a fast-charging strategy without lithium-plating due to negative voltages of $U_{neg}$. This objective function is subject to side conditions as the minimum desired voltage of the negative electrode ($U_{neg,min}$) by Equation (9), maximum desired current $I_{fast,max}$ by Equation (10) and maximum desired charge amount $Q_{max}$ by Equation (11) [52].

$$maximize\ I_{fast}(t) = \left(U_{ocv,neg}(t) - U_{neg}(t) - U_{RC1,neg}(t) - U_{RC2,neg}(t)\right)/R0_{neg}(t) \tag{8}$$

subject to:

$$U_{neg}(t) \ \geq \ U_{neg,min} \tag{9}$$

$$I_{fast}(t) \ \leq \ I_{fast,max} \tag{10}$$

$$Q_{max} \geq \int I_{fast}(t)\, dt \tag{11}$$

### 2.2. Model Parameterization and Validation

Commercial 3-electrode cells in coin cell setup (PAT-Cells from EL-Cell GmbH, Hamburg, Germany) were used for the model parameterization. The cell configuration is listed in Table 1.

The cells had a defined SOC of 0% at 2.9 V while 100% SOC was defined at 4.1 V open-circuit voltage. The minimum and maximum allowed cell voltage under current load were 2.5 V and 4.2 V, respectively. All experimental tests were conducted by a battery-test system (PAT-Tester-x-8 from EL-Cell GmbH) in a climate chamber (WT11-600/40 from Weiss Technik GmbH, Reiskirchen, Germany) at 20 °C.

The EECM equations from Section 2.1 were implemented in MATLAB/Simulink. All EECM parameters are Coulombic SOC dependent and represented by 1D-lookup tables for the constant temperature of 20 °C. In this section, a multi-step parameterization approach

is presented and illustrated in Figure 2. This method enabled precise simulation fidelity of the EECM, as proved by the validation of different current profiles later on.

**Table 1.** Cell configuration.

|  | Positive Electrode | Negative Electrode |
|---|---|---|
| **active material** | NMC622 | SMG-A5 |
| **current collector** | 20 μm aluminum | 10 μm copper |
| **coating thickness** | 68 μm | 82 μm |
| **calendered coating density** | 3 g/cm$^3$ | 1.3 g/cm$^3$ |
| **cross-sectional area** | 2.54 cm$^2$ | 2.54 cm$^2$ |
| **active material percentage** | 95.5% | 93.0% |
| **theoretical areal capacity** | 3.41 mAh/cm$^2$ | 3.60 mAh/cm$^2$ |
| **electrolyte** | 100 μL 1.0M LiPF6 in EC:EMC (3:7) + 2% VC | |
| **separator** | 260 μm Whatman GF/A with lithium reference ring | |

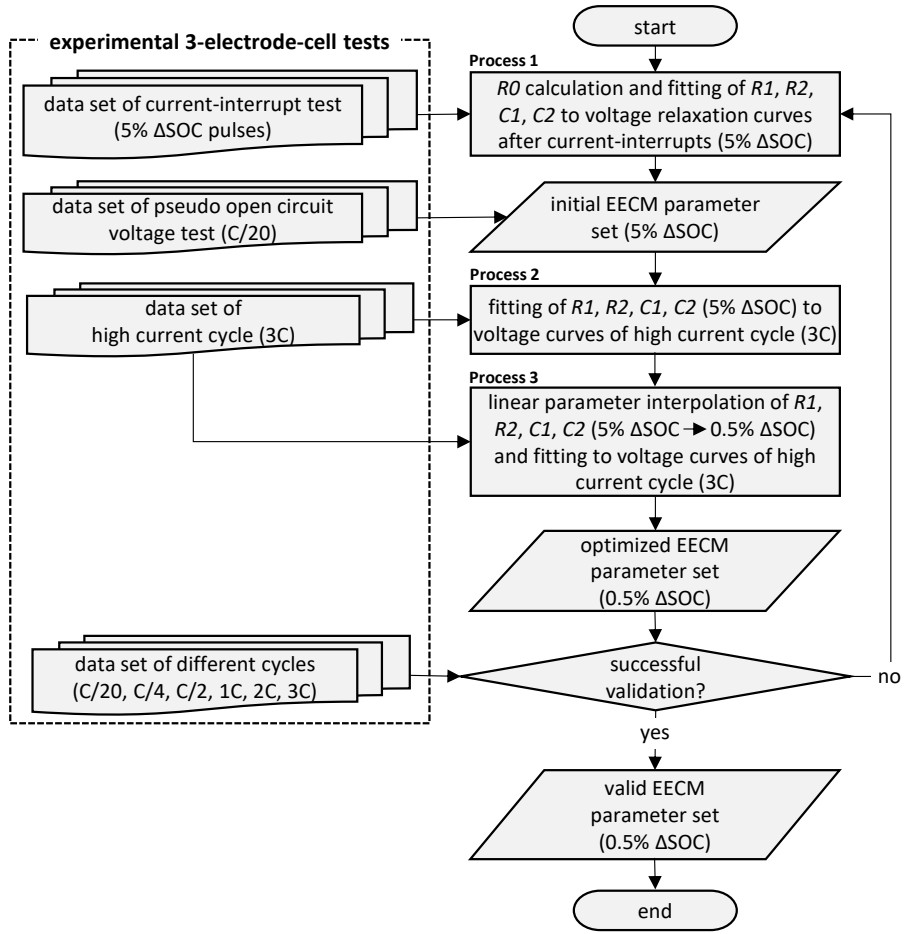

**Figure 2.** Flow chart of the novel Electrode Equivalent Circuit Model (EECM) parameterization approach with two *RC* branches for every electrode. Different experimental tests with 3-electrode cells are required for the parameterization. The three-step fitting process with parameter interpolation enables a fine SOC resolution for precise simulation fidelity. The values in brackets are the specifications of the applied parameterization for this study.

**Process 1**: As illustrated in Figure 2, this parameterization approach needs different experimental data from 3-electrode cells, with a lithium reference electrode between the negative and positive electrodes. For this study, the experimental current-interrupt test started at 0% SOC and was subdivided by 0.5C charge pulses (5% ΔSOC), which were followed by current-less relaxation phases of 1 h, respectively. The Coulombic SOC of the

cell does not change during the current-less relaxation phase, which enables the Coulombic SOC-dependent determination of the parameters. The instantaneous change of the electrode voltages after stopping the current pulse for the relaxation at different SOC allowed the calculation of the ohmic resistances $R0_{neg}$(SOC) and $R0_{pos}$(SOC) by Ohm's Law. The specific electrodes' $R1$(SOC), $R2$(SOC), $C1$(SOC) and $C2$(SOC) were obtained by fitting the measured electrode voltage curves during relaxation to the exponential equations of the $RC$ branches. This fitting procedure (see Process 1 in Figure 2) is performed by the MATLAB Curve Fitting Toolbox [54] using the nonlinear least-squares method with Trust-Region algorithm and was already described in detail in our previous study [52].

The new parameterization approach is an enhanced method that enables a much higher SOC resolution than 5%, in order to increase the simulation fidelity. Experimental C/20 voltage profiles of the electrodes were used as pseudo open circuit voltage (ocv) curves for $U_{ocv,neg}$ and $U_{ocv,pos}$. The measured pseudo ocv curves, the calculated ohmic resistances and the fitted parameters from seven tested 3-electrode cells completed the initial EECM parameter set after Process 1 (see Figure 2). This parameter set is depicted as solid lines with mean values and standard deviations in Figure 3.

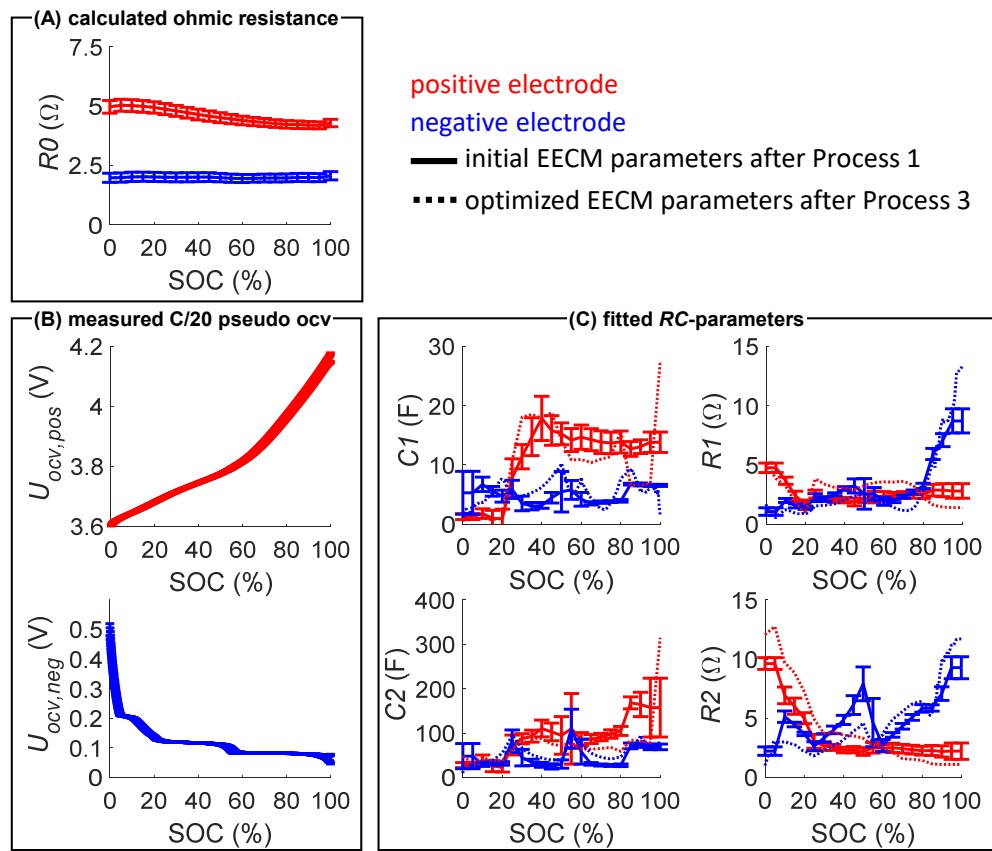

**Figure 3.** Determined parameter set of the Electrode Equivalent Circuit Model (EECM) for the given cell configuration. The initial parameter set is obtained after Process 1. (**A**) The ohmic resistance $R0$ is calculated by Ohm's Law based on the current-interrupt test. (**B**) The pseudo open circuit voltage (ocv) curves ($U_{ocv,pos}$ and $U_{ocv,neg}$) are received by the C/20 measurement. The voltages of both electrodes are measured versus the lithium reference electrode. (**C**) The parameters of the $RC$ branches are obtained by the fitting processes (Process 1 to Process 3). The optimized parameter set is obtained after Process 3 and is the basis of the validation. Besides $R0$, all parameters show strong SOC dependency.

**Process 2**: After the initial parameter set was determined, the parameters for the electrodes' $R1$(SOC), $R2$(SOC), $C1$(SOC) and $C2$(SOC) were fitted in MATLAB/Simulink using the nonlinear least-squares method and Trust-Region Reflective algorithm by the Parameter

Estimation application [55]. The best result was obtained when the *RC* parameters were fitted to the experimental cycle with the highest current (3C cycle for this study), as higher currents induce higher *RC*-based overpotentials. High *RC*-based overpotentials are more suitable for the fitting, as the impact of measurement noise is reduced. The pseudo ocv curves were not fitted, as they already have sufficient precision based on the extraction from the C/20 curves. Additionally, the *R0*(SOC) parameters also were not fitted, as they can be calculated precisely by Ohm's Law and only showed minor differences between a ΔSOC of 5%.

**Process 3**: After fitting the parameters in Process 2, all EECM parameters were linearly interpolated to a higher SOC resolution (0.5% ΔSOC) using the function interp1 [56] in MATLAB. Then the interpolated parameters for electrodes *R1*(SOC), *R2*(SOC), *C1*(SOC) and *C2*(SOC) were fitted once again to the same high current charge cycle from before. This last fitting process ends up in the optimized EECM parameter set, as depicted as dotted lines in Figure 3.

Subsequent to Process 3, the optimized parameter set was used for the validation of different charge cycles, in order to prove if the optimized EECM parameters result in a valid simulation fidelity. The validation shows good alignment of the simulation and the experimental data of the seven 3-electrode cells as depicted in Figure 4.

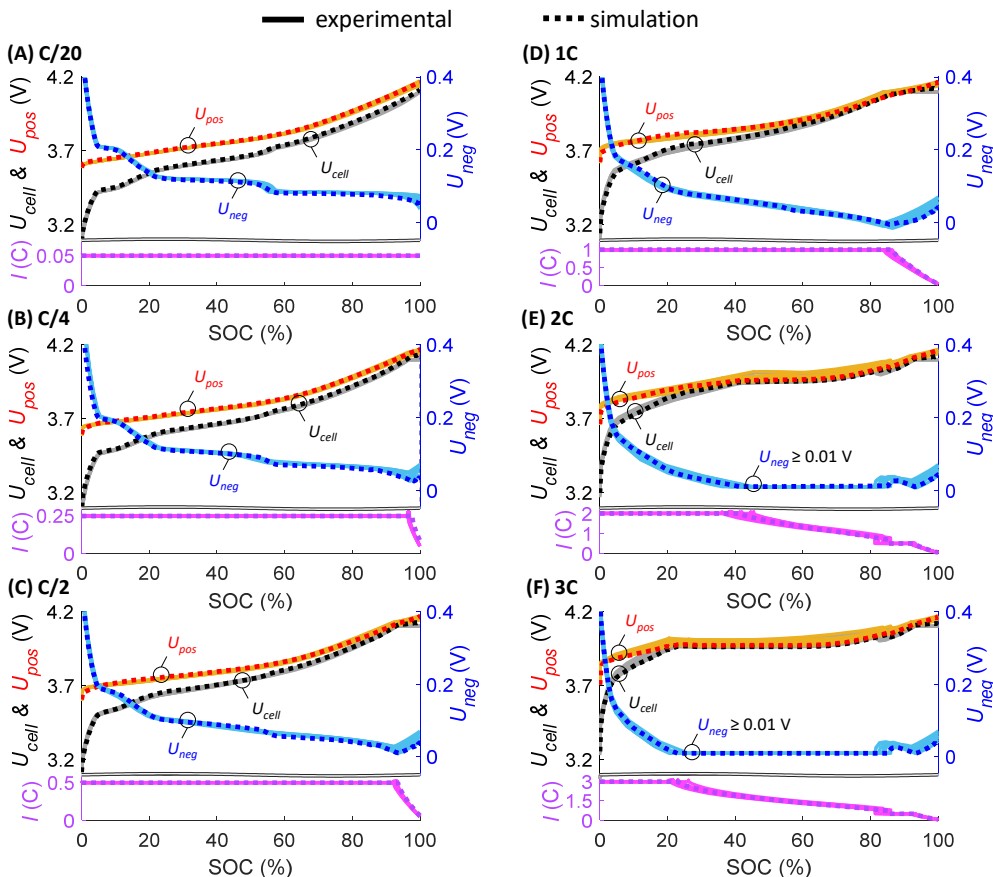

**Figure 4.** Validation of different experimental charge cycles of seven 3-electrode-cells and the simulation data based on the optimized model parameters. The voltages of both electrodes ($U_{pos}$ and $U_{neg}$) are measured versus the lithium reference electrode, whereas the cell voltage ($U_{cell}$) is the differential voltage of the electrodes. (**A**) C/20 CC charge. (**B**) C/4 CCCV (CV at 4.1 V $U_{cell}$ until I < C/20). (**C**) C/2 CCCV (CV at 4.1 V $U_{cell}$ until I < C/20). (**D**) 1C CCCV (CV at 4.1 V $U_{cell}$ until I < C/20). (**E**) 2C CC; CV at 0.01 V $U_{neg}$ until 80% SOC, then C/2 CCCV (CV at 4.1 V $U_{cell}$ until I < C/20). (**F**) 3C CC; CV at 0.01V $U_{neg}$ until 80% SOC, then C/2 CCCV (CV at 4.1 V $U_{cell}$ until I < C/20). The charge cycles (**E**,**F**) have online controlled CV charge cycles for the negative electrode, in order to prevent lithium-plating due to high negative potentials.

With regard to the seven tested cells and six charge cycles, the root mean square error (RMSE) for $U_{cell}$ ranges between 4–20 mV, whereas $U_{pos}$ ranges between 8–20 mV and $U_{neg}$ ranges between 1–6 mV. These slight deviations represent a successful validation, hence the optimized parameter set (see Figure 3) is valid for 20 °C, 0–100% SOC and currents less than or equal to 3C.

## 3. Results

### 3.1. Model-Based Design of Different Fast-Charging Strategies

State of the art CCCV fast-charging strategies do not consider the voltage of the negative electrode, which may lead to lithium-plating if this voltage drops below 0 V. Based on Equations (8)–(11) in Section 2.1, the validated EECM was used for computing a fast-charging strategy for 0–80% SOC with a maximum current of 3C ($I_{fast,max}$). The minimum allowed voltage of the negative electrode ($U_{neg,min}$) was set to 10 mV, in order to prevent lithium-plating while having a safety puffer based on the determined maximum RMSE of 6 mV for $U_{neg}$. These constraints were used for the simulation of the optimized fast-charging strategy, denoted as 3C(EECM-opt). Figure 5 contrasts 3C(EECM-opt) versus a state-of-the-art 3C CCCV fast-charging strategy until 80% SOC, denoted as 3C(CCCV).

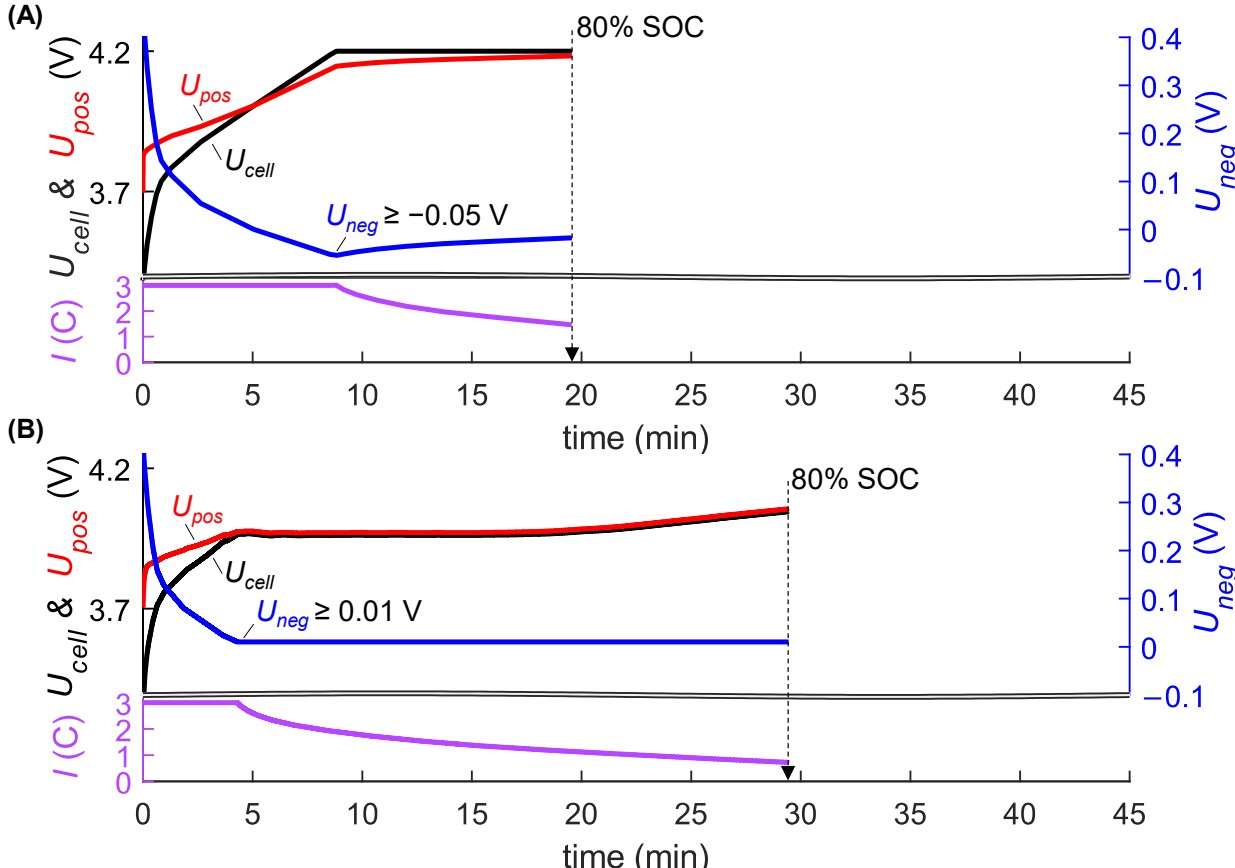

**Figure 5.** Comparison of the different fast-charging strategies and associated voltage curves. The voltages of both electrodes ($U_{pos}$ and $U_{neg}$) are measured versus the lithium reference electrode, whereas the cell voltage ($U_{cell}$) is the differential voltage of the electrodes. (**A**) The state-of-the-art fast-charging strategy 3C(CCCV) leads to negative voltages of the negative electrode. (**B**) Optimized fast-charging strategy 3C(EECM-opt) with 0.01 V minimum allowed voltage of the negative electrode. The 3C(EECM-opt) prevents negative voltages of the negative electrode, in order to prevent lithium-plating.

The state-of-the-art fast-charging strategy 3C(CCCV) results in a charging time of about 20 min until 80% SOC but leads to negative voltages of the negative electrode down to −0.05 V at the end of the CC-phase. The optimized fast-charging strategy 3C(EECM-

opt) needs about 29 min until 80% SOC. The charging time range of the two strategies between 20–29 min for 0–80% SOC is in good accordance with other minimum fast-charging durations for lithium-ion batteries from the literature [3,6].

### 3.2. Experimental Fast-Charging Cycling and Assessment

In order to assess the aging impact, the two different fast-charging strategies were used for experimental cycling. Therefore, the 3-electrode cells were cycled with the two different fast-charging strategies. The state-of-the-art 3C(CCCV) fast-charging strategy was programmed in the battery-test system by an online cell voltage control of the battery tester. As soon as the cell voltage reached 4.2 V with the 3C CC step, the CV step was performed until 80% SOC. In contrast to that, the optimized strategy 3C(EECM-opt) was implemented in the battery-test system by a current profile lookup-table that was derived from the EECM simulation before the cycling started. After finishing the charging until 80% SOC, both charging strategies had a 10 min pause, followed by a 1C, CCCV (I < C/20) discharge to 2.9 V (0% SOC) and another 10 min pause prior to the next fast-charging cycle. All fast-charging cycles during cycling were limited to 133 mAh/$g_{NMC}$ which equals 80% SOC of 166 mAh/$g_{NMC}$ for a total 0.5C CCCV discharge between 4.1 V and 2.9 V (100–0% SOC) before the cycling was started.

Optical post-mortem analyses were carried out for selected cells at 0% SOC before cycling and after 100 cycles as well as after 300 cycles, in order to inspect the negative electrodes for lithium-plating. The post-mortem analyses were conducted by choosing one representative cell per strategy, which was disassembled under argon atmosphere inside a glovebox with less or equal to 0.5 ppm $H_2O$ and $O_2$. Therefore, the lid of the coin cell (PAT-Cells from EL-Cell GmbH) was unscrewed and the negative electrode was removed with tweezers. Afterward, the negative electrode was stored for at least two hours inside the glovebox, in order to dry the electrolyte. In order to investigate the electrode sample outside the glovebox with a microscope, the sample was placed inside a hermetically sealed sample holder. The self-made sample holder places the electrode sample inside an O-ring seal that is pressed by a cover glass between two screwed steel plates whereas the upper plate has a recess for the cover glass. The electrode sample was imaged through the glass cover by a digital microscope (VHX-7000 from Keyence Corporation, Osaka, Japan). Figure 6 shows the pictures of the selected negative electrodes before and after cycling with the two different fast-charging strategies.

The fast-charging strategy 3C(CCCV) led to a significant amount of lithium-plating during the first 100 fast-charging cycles. The lithium-plating was partly connected with separator fibers. Lithium-plating often appears as sharp dendrites [3], which explains the interweaving with the separator fibers. In contrast, the optimized fast-charging strategy 3C(EECM-opt) prevented lithium-plating. Even after 300 cycles, there was no visible lithium-plating by charging with 3C(EECM-opt). The color of the graphite changed between 100 and 300 cycles. This could be explained by the fact that the color of the negative electrode is dependent on the degree of lithiation [57], which changes during aging due to the loss of lithium-ions. However, the color change is no sign of lithium-plating as the negative electrode still showed the same surface morphology of the graphite. We also checked the NMC electrodes by the microscope with 100× zoom but did not see any differences between the both methods and different cycle counts. For analyzing possible degradation effects of NMC particles, a much higher magnification is necessary [10], which was not possible to investigate in the framework of this study.

Before and during the cycling, the cells were periodically characterized with different discharge rate tests from 0.5C CC to 3C CC at different cycle counts (0, 50, 100, 200 and 300 cycles) in order to determine the capacity retention. The specific discharge capacity retention after different cycle counts and different current rates are depicted in Figure 7A–D, whereas Figure 7E illustrates the course of the different fast-charging durations of 3C(CCCV) and 3C(EECM-opt).

Before the fast-charging cycling started, the cells had about 149 mAh/$g_{NMC}$ capacity for a full 0.5C CC discharge. As illustrated in Figure 7A, the cells with 3C(CCCV) dropped to $124 \pm 7$ mAh/$g_{NMC}$ (12–21% loss) after 100 cycles whereas the cells with 3C(EECM-opt) only dropped to $146 \pm 1$ mAh/$g_{NMC}$ (1–3% loss). The cycling of the cells with 3C(CCCV) was stopped after 100 cycles due to significant degradation as some cells already had a 20% capacity fade. The cells with 3C(EECM-opt) showed a linear aging behavior and only lost about 2% per 100 cycles, which would result in an extrapolated cycle life of 1000 fast-charging cycles until 20% capacity fade with 0.5C discharges, respectively.

As depicted in Figure 7B–D, when higher currents were applied, the gap in discharge capacity retention between both strategies decreased during the cycling. This could be explained by lithium-stripping during discharging. Lithium-stripping is the recovering effect of some plated lithium to reusable lithium-ions during discharging and the stripping rate increases with higher currents [3].

**Figure 6.** Optical post-mortem inspection (100× zoom) of the negative electrodes under argon atmosphere. (**A**) There is no visible lithium-plating before cycling. (**B**) The 3C(CCCV) strategy leads to lithium-plating during the first 100 cycles, which is partly attached to separator fibers. (**C**) The optimized strategy 3C(EECM-opt) does not lead to visible lithium-plating after 100 cycles. (**D**) The optimized strategy 3C(EECM-opt) does not lead to visible lithium-plating after 300 cycles.

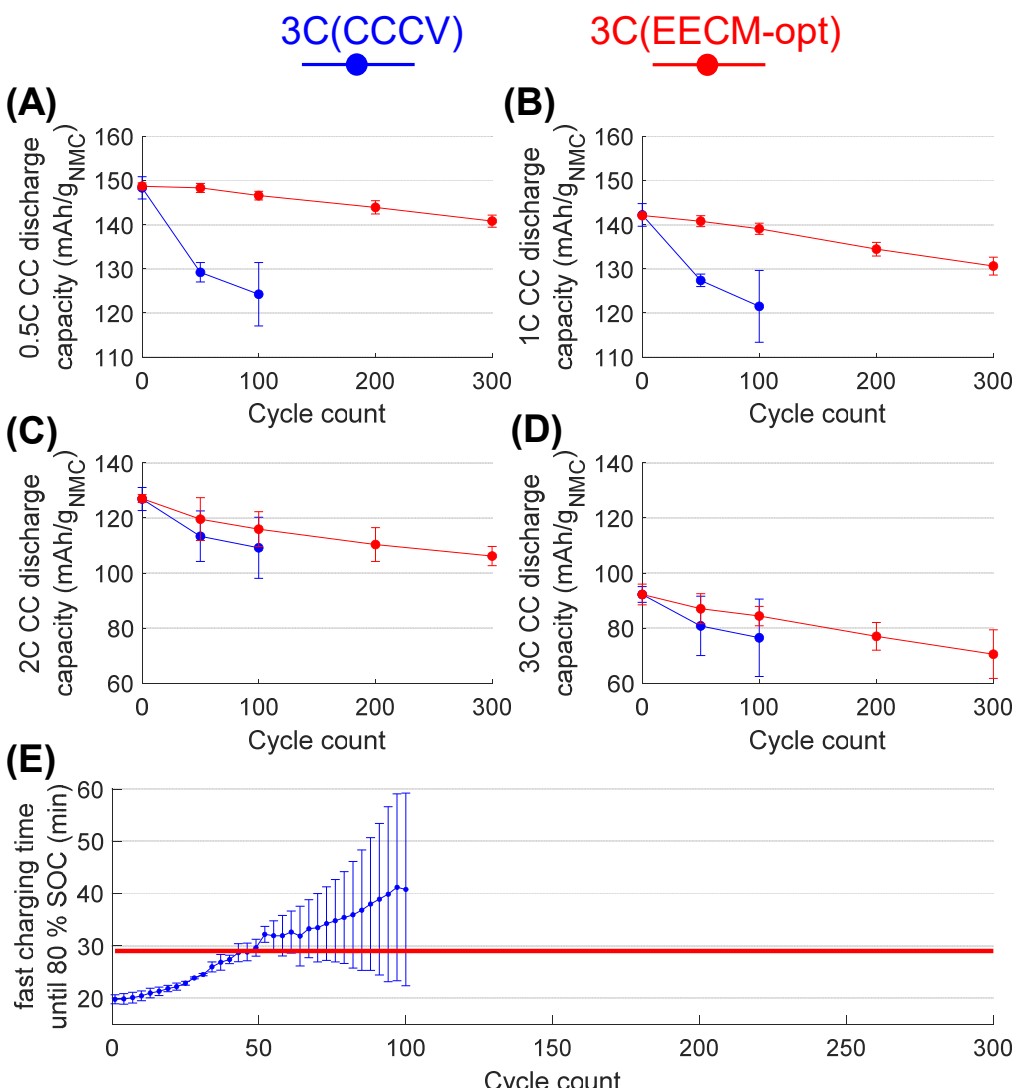

**Figure 7.** Specific discharge capacity retention and fast-charging time during cycling with the two different fast-charging strategies. (**A**) During cycling, the state-of-the-art 3C(CCCV) fast-charging strategy leads to significantly faster 0.5C (**A**) and 1C (**B**) discharge capacity loss compared to 3C(EECM-opt). The 2C (**C**) and 3C (**D**) discharge capacity does not differ significantly during cycling between both strategies. (**E**) During cycling, the fast-charging time of 3C(CCCV) increases whereas the duration of 3C(EECM-opt) is constant for 300 cycles.

In Figure 7E, the mean charging time of 3C(CCCV) increased to about 30 min after 50 cycles and passed the constant fast-charging time of 29 min for 3C(EECM-opt). The increased fast-charging time of 3C(CCCV) is related to increased cell resistances, as the CC phase until 4.2 V ended earlier so that the CV phase lasted longer, which resulted in longer charging times until 80% SOC. The optimized fast-charging strategy 3C(EECM-opt) had no change in fast-charging speed as the maximum cell voltage was always below the maximum allowed voltage of 4.2 V. The maximum cell voltage for 3C(EECM-opt) during the first fast-charging cycle was 4.048 V $\pm$ 0.019 V, whereas the maximum cell voltage in the 300th cycle was 4.118 V $\pm$ 0.018 V due to also increased cell resistances.

The different aging performances can be correlated with the diagnosis of lithium-plating for 3C(CCCV), as already shown in Figure 6B. The lithium-plating reduced the capacity significantly and led to wider spreads of the standard deviation, as lithium-plating evolves irregularly.

In order to calculate the internal resistance during the cycling, current-interrupt tests were conducted before cycling and after 100 and 300 fast-charging cycles. Therefore, a 0.5C CC current-interrupt test was performed between 5–95% SOC in 5% $\Delta$SOC steps, based on a prior full-cycle capacity test. Equation (12) calculates the total internal resistance $R$ by Ohm's Law based on the total voltage difference $\Delta U_{relax}$ during the 1 h relaxation phase after the current pulse $I_{pulse}$ was stopped. The calculated resistance $R$ was transformed into the specific resistances $\rho$ based on the electrode's cross-sectional area $A$ and coating thickness $L$ of the electrodes from Table 1 by Equations (13)–(15).

$$R = \Delta U / \Delta I = \Delta U_{relax} / I_{pulse} \tag{12}$$

$$\rho_{cell} = R_{cell} * A / (L_{pos} + L_{neg}) \tag{13}$$

$$\rho_{pos} = R_{pos} * A / L_{cat} \tag{14}$$

$$\rho_{neg} = R_{neg} * A / L_{neg} \tag{15}$$

Figure 8 illustrates the specific resistances $\rho$ over the charged capacity during the current-interrupt test, between 5–95% SOC at different cycle counts for both fast-charging strategies.

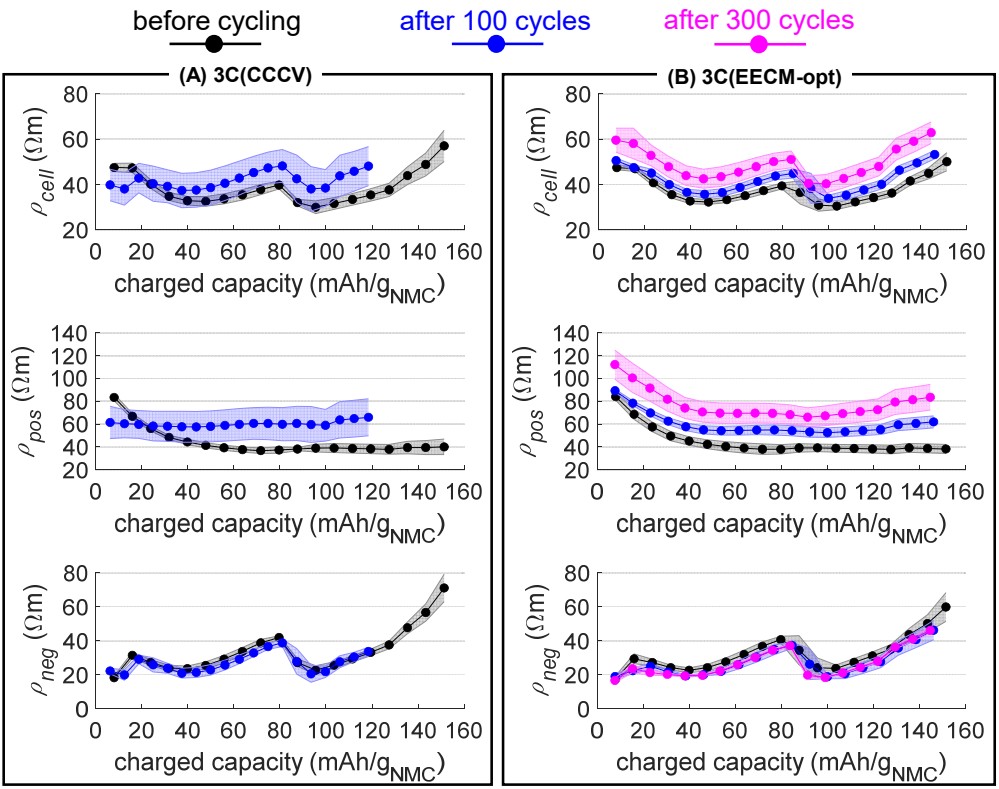

**Figure 8.** Specific cell and electrode resistances at different cycle counts between 5–95% SOC. (**A**) specific cell and electrode resistances for 3C(CCCV) cycling. (**B**) specific cell and electrode resistances for 3C(EECM-opt) cycling. The cycling of both fast-charging strategies leads to increased positive electrode resistances whereas the negative electrode resistances do not change significantly —even for 3C(CCCV), which results in lithium-plating.

The specific resistances before cycling (black lines) were identical for both fast-charging strategies, as there were no differences in the prior test protocols. The specific resistance graphs after 100 cycles (blue lines) and 300 cycles (magenta lines) are getting shorter, as the maximum capacity retention decreased during aging. Therefore, the graphs were plotted over the charged capacity instead of the SOC percentage as this would lead to a distortion of the trends.

Both strategies led to overall increased $\rho_{pos}$ during aging except during the first 30 mAh/g$_{NMC}$ of 3C(CCCV) after 100 cycles. The reduced $\rho_{pos}$(100 cycles) for 3C(CCCV) during the first 30 mAh/g$_{NMC}$ could be explained by fractured NMC particles near the separator, due to too high current densities. During charging, these particles initially undergo a deintercalation of lithium-ions. Combined with the fact that fractured particles can lead to better charge transfer kinetics due to higher active material surface area in contact with the electrolyte [58], this may explain the decreased $\rho_{pos}$ for the 3C(CCCV) strategy during the first 30 mAh/g$_{NMC}$ at higher cycle counts. Beyond 30 mAh/g$_{NMC}$, $\rho_{pos}$(100 cycles) for both strategies was almost constant at about 60 Ωm which is about 50% higher compared to $\rho_{pos}$(0 cycles). The $\rho_{pos}$(300 cycles) for 3C(EECM-opt) increased up to 70–80 Ωm beyond 30 mAh/g$_{NMC}$.

The increased $\rho_{pos}$ can be associated with different degradation processes, such as completely fractured NMC particles, or growth of the positive electrode–electrolyte interphase (pSEI) layer. Completely fractured NMC particles may isolate from the solid electrode and lengthen the ionic pathways, which leads to increased resistance. Moreover, a loss of electrical conductivity to the positive electrode will result in active material loss, and thus capacity loss. Another related degradation process is the pSEI growth during aging, which is accelerated by particle fractures and cracks, as this evokes new pSEI formation and will increase the positive electrode resistance [10]. However, we did not prove if some or all of these possible phenomena occurred, as it is was not possible with our equipment.

In contrast to the $\rho_{pos}$, the $\rho_{neg}$ did not change significantly—although 3C(CCCV) led to significant lithium-plating, as already depicted in Figure 6B. This means that the lithium-plating on the negative electrode does not negatively affect the intercalation process during charging. However, the lithium-plating still led to lower capacity retention. SEI growth on the negative electrode is another relevant aging process associated with increased resistances [59]. As the $\rho_{neg}$ did not change significantly, it is assumed that there was no major SEI growth.

The increased $\rho_{cell}$ at higher cycle counts are only related to the increased $\rho_{pos}$ as $\rho_{neg}$ did not change significantly. Increased cell resistances decrease the CC duration as this phase ends earlier if the maximum cell voltage is reached. This means that only $\rho_{pos}$ is correlating with the increased fast-charging time during cycling of 3C(CCCV) as already illustrated in Figure 7B.

The determination of the specific cell resistance allows the upscaling comparison of 3-electrode coin cells to large format cells with the same electrode and electrolyte type. In our previous study, we showed that 3-electrode coin cells and multi-layer pouch-cells had similar specific resistances, as both cell types led to similar overpotentials when using the same C-rate [52]. Thus, the presented EECM method in this paper of determining an optimized fast-charging strategy for 3-electrode cells, as C-rate profile, is suitable to be upscaled to pouch cells, if the specific resistance is similar. However, we were not able to produce pouch cells in the framework of this study, as this is part of future investigations.

## 4. Conclusions

The design of durable fast-charging strategies for lithium-ion batteries that prevent rapid cell degradation is important to guarantee high performance, safe conditions and long cycle life. Lithium-plating is the major degradation effect related to fast charging, as it causes fast cell degradation and, in the worst case, a sudden cell defect due to an internal short circuit. Conventional CCCV fast charging based on the cell voltage does not consider lithium-plating, whereas variable current profiles by model-based control of the negative electrode's voltage may prevent lithium-plating. In this study, the Electrode Equivalent Circuit Model (EECM) from prior research was advanced, in order to compute an optimized fast-charging strategy that prevents lithium-plating by negative potentials of the negative electrode. A new multi-step fitting parameterization method was developed that resulted in high simulation fidelity.

The EECM was used in order to compute an optimized fast-charging strategy with 3C maximum current and 10 mV minimum allowed voltage of the negative electrode in order to prevent lithium-plating. The profile lasts 29 min for 0–80% SOC and was applied to the cycling of NMC622/G cells. Identical cells were cycled with conventional 3C CCCV with 20 min for 0–80% SOC at the beginning of the cycling. This strategy led to severe degradation of 20% capacity fade after 100 cycles due to heavy lithium-plating, detected by optical microscopy. The optimized 3C fast-charging strategy only led to linear degradation of about 2% capacity fade per 100 cycles in the first 300 cycles and did not result in any visible lithium-plating under the microscope. Based on a linear extrapolation until 20% capacity fade, about 1000 fast-charging cycles are expectable. This cycle life is sufficient for most applications in the automotive or consumer electronics industry.

The change of the resistances during cycling was analyzed by current-interrupt tests. It was found that the negative electrode's resistance did not change significantly for both fast-charging strategies, although the conventional 3C CCCV evoked lithium-plating. The positive electrode's resistance increased significantly, which led to an increased charging time for the conventional 3C CCCV due to shorter CC phases. The mean fast-charging time for 0–80% SOC of the conventional 3C CCCV strategy increased linearly from 20 min in the first cycle to 40 min in the 100th cycle. This concludes that the degradation of the positive electrode is the main limiting factor to charging speed limitations during cycling. In contrast, the optimized 3C fast-charging method for 0–80% SOC had a constant charging duration of 29 min for all tested 300 cycles, as the maximum cell voltage was always below the maximum allowed voltage of 4.2 V. In summary, by means of the EECM, it is possible to design a durable fast-charging strategy without lithium-plating, which leads to significantly better capacity retention compared to conventional CCCV fast charging and also allows for shorter mean charging times in the course of aging.

**Author Contributions:** R.D.: conceptualization, methodology, software, validation, data curation, writing—original draft preparation, writing—review and editing, visualization. F.L.: writing—review and editing. M.K.: writing—review and editing, supervision. All authors have read and agreed to the published version of the manuscript.

**Funding:** This work was supported by the project "FastChargeLongLife" (03XP0313) as part of the competence cluster "BattNutzung" by the Federal Ministry of Education and Research in Germany (BMBF). Furthermore, we acknowledge support from the German Research Foundation and the Open Access Publication Funds of Technische Universität Braunschweig.

**Data Availability Statement:** The data are not publicly available because the data are part of other unpublished studies.

**Acknowledgments:** Additional gratitude belongs to the project partners from the Institute for Particle Technology (iPAT) of the TU Braunschweig and the Institute of Energy and Process Systems Engineering (iNeS) of the TU Braunschweig. Sincere thanks are given to Fabian Katschewitz for supporting the battery tests.

**Conflicts of Interest:** The authors declare no conflict of interest.

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
