# Peer review of "Durable Fast Charging of Lithium-Ion Batteries Based on Simulations with an Electrode Equivalent Circuit Model"

_batteries, doi:10.3390/batteries8040030_

Round 1

Reviewer 1 Report

This manuscript described a new simulation method, the Electrode Equivalent Circuit Model (EECM), to optimize the fast charging of Li-ion batteries up to 3C. The authors used the experimental datasets for parameterization which feeds back to their EECM model. The manuscript was well written, and the results are sound. This paper proposed the EECM approach to mitigate the common lithium-plating issue at the interface (which the CCCV method cannot avoid), therefore prolonging the cell lifetime. It is worth publishing subject to a minor revision:

  • The details of post-mortem characterisation should be added to the Experiment Section.
  • The authors used commercial 3-electrode-cells for investigation, how about the fast-charging application of EECM in pouch type?
  • The authors demonstrated microstructure at anode surface using both EECM and CCCV methods. Are there any impacts on the NMC cathode side by using the EECM?

Reviewer 2 Report

This article develops an advanced electrode equivalent circuit model based on a 3-electrode-cell in order to simulate fast charging strategies without lithium plating. Compared the conventional 3C CCCV, this strategy can achieve an optimized fast charging strategy without lithium-plating was simulated that lasted about 29 min for 0–80% SOC with a maximum current of 3C. In addition, this strategy shows slow capacity fade and better duration. This is a comprehensive research work. and I believe it is suitable for publication in batteries. Still, there are some points I'm confused, and I think the author should make revisions to achieve better expressions for general readers:

  1. Some abbreviations, such as SOC, 3C, and 3C CCCV, were shown in the abstract, to make it easier for readers to read, try to avoid use abbreviations in the abstract.
  2. The function of article keywords is mainly to quickly convey the content of research work to readers, but “optimization” is only the result of work, and it is not very suitable to appear as a keyword.
  3. In this article, author think the electrode and electrolyte properties or cell configuration will influence the charging capacity, please explain why and how these elements affect the charging process.
  4. The lithium ions plating maybe relate with the transport of lithium ions, please refer to the following article for reference.
  • Oriented UiO-67 Metal–Organic Framework Membrane with Fast and Selective Lithium-Ion Transport, Angew. Chem. Int. Ed. 2021, 60, e202115443
  • Super-assembly of freestanding graphene oxide-aramid fiber membrane with T-mode subnanochannels for sensitive ion transport, Analyst, 2022,147, 652-660
  • Enhanced Ion Sieving of Graphene Oxide Membranes via Surface Amine Functionalization, J. Am. Chem. Soc. 2021, 143, 5080−5090
  • Interfacial Super-Assembly of Ordered Mesoporous Carbon-Silica/AAO Hybrid Membrane with Enhanced Permselectivity for Temperature- and pH-Sensitive Smart Ion Transport, Angew. Chem. Int. Ed. 2021, 60, 26167-26176
  1. It would be better if a mechanism explanation diagram was replenished in this article to illustrate why this model and equipment can achieve optimization performance.

Reviewer 3 Report

In this work, the authors developed an advanced electrode equivalent circuit model in order to simulate fast charging strategies without lithium-plating. The experiments showed that the optimized strategy prevented lithium-plating and led to 2% capacity fade every 100 fast charging cycles. In contrast, the conventional strategy led to lithium-plating, about 20% capacity fade after 100 fast charging cycles and the fast charging duration extended from 20 min to over 30 min due to increased cell resistances. The results are interesting and good, and the topic lies within the scope of the journal, the manuscript can be accepted, but there are some issues that the authors can address.

  1. Please show the explication of acronyms mentioned in the manuscript, for example, State of charge (SOC), Constant Current, Constant Voltage (CCCV). 
  2. The glass fiber can't be utilized in large format cells for real applications. Why the authors chose GF instead of PP or PE separators?
  3. As authors said, 'Completely fractured NMC particles may isolate from the solid electrode and lengthen the ionic pathways, which leads to an increased resistance.' Authors should provide SEM or other characterizations to further prove it.
  4. A more detailed data figure of cycling performance with and without EECM can be added.  

Reviewer 4 Report

  1. There are other references that the authors can look into for plating detection and its effect on battery performance using a three-electrode configuration [1,2]. Although the references do not focus on modeling or high charge scenarios, it is pertinent to discuss the literature relevant to the idea being presented and argued. Also, there are single-particle models combined with ECM formalism to provide electrode states as a function of SOC (state-of-charge) and SOH (state-of-health) [3,4].
  2. Another point for the authors is that since the ECM or SPM often ignores the potential of the electrolyte, this can also cause the negative electrode to go below 0V and lead to lithium plating conditions.
  3. In Figures 4 and 5, it would be useful to mention that Upos and Uneg are measured w.r.t lithium while the Ucell is the differential voltage between high and low voltage electrodes.
  4. How was the fidelity of the parameters tested, since RC time constants could be interchanged since this is the optimization of essentially 8 parameters?
  5. Could the authors comment on how the model captures the negative voltage, is it due to the fitting of the OCV or from the RC parameters? This is unclear because the authors are measuring the individual electrode voltages as well.
  6. The authors mention that the relaxation voltage does not change the SOC, they may want to be explicit that they mean Coulombic SOC and not true SOC (thermodynamic state of the electrode, which would change)
  7. The authors' cursory mention that they use experimental OCVs, where they fitted to a formalism such as Redlich-Kister or a derivative of the Fermi-Dirac distribution, or the OCV data used as a LUT?

References:

  1. S.P. Rangarajan, S. Sarkar, Y. Barsukov, P.P. Mukherjee, 3ε–A Versatile Operando Analytics Toolbox in Energy Storage, ACS Omega. (2021) acsomega.1c05494. https://doi.org/10.1021/acsomega.1c05494.
  2. S. Perumaram Rangarajan, Y. Barsukov, P.P. Mukherjee, Plating energy as a universal descriptor to classify accelerated cell failure under operational extremes, Cell Reports Physical Science. (2022) 100720. https://doi.org/10.1016/j.xcrp.2021.100720.
  3. M. Daigle, C.S. Kulkarni, Electrochemistry-based Battery Modeling for Prognostics, in: Annual Conference of the Prognostics and Health Management Society, 2013: p. 13.
  4. M. Daigle, C.S. Kulkarni, End-of-discharge and End-of-life Prediction in Lithium-ion Batteries with Electrochemistry-based Aging Models, in: AIAA Infotech @ Aerospace, American Institute of Aeronautics and Astronautics, San Diego, California, USA, 2016. https://doi.org/10.2514/6.2016-2132.
